# Tetrapod limb and sarcopterygian fin regeneration share a core genetic programme

Acacio F. Nogueira[1,*], Carinne M. Costa[1,*], Jamily Lorena[1], Rodrigo N. Moreira[1], Gabriela N. Frota-Lima[1], Carolina Furtado[2], Mark Robinson[3], Chris T. Amemiya[3,4], Sylvain Darnet[1] & Igor Schneider[1]

Salamanders are the only living tetrapods capable of fully regenerating limbs. The discovery of salamander lineage-specific genes (LSGs) expressed during limb regeneration suggests that this capacity is a salamander novelty. Conversely, recent paleontological evidence supports a deeper evolutionary origin, before the occurrence of salamanders in the fossil record. Here we show that lungfishes, the sister group of tetrapods, regenerate their fins through morphological steps equivalent to those seen in salamanders. Lungfish *de novo* transcriptome assembly and differential gene expression analysis reveal notable parallels between lungfish and salamander appendage regeneration, including strong downregulation of muscle proteins and upregulation of oncogenes, developmental genes and lungfish LSGs. MARCKS-like protein (MLP), recently discovered as a regeneration-initiating molecule in salamander, is likewise upregulated during early stages of lungfish fin regeneration. Taken together, our results lend strong support for the hypothesis that tetrapods inherited a *bona fide* limb regeneration programme concomitant with the fin-to-limb transition.

[1] Instituto de Ciências Biológicas, Universidade Federal do Pará, Rua Augusto Correa, 01, Belém 66075-110, Brazil. [2] Unidade Genômica, Programa de Genética, Instituto Nacional do Câncer, Rio de Janeiro 20230-240, Brazil. [3] Benaroya Research Institute at Virginia Mason, 1201 Ninth Avenue, Seattle, Washington 98101, USA. [4] Department of Biology, University of Washington 106 Kincaid, Seattle, Washington 98195, USA. * These authors contributed equally to this work. Correspondence and requests for materials should be addressed to I.S. (email: ischneider@ufpa.br).

The question of why urodele amphibians are the only tetrapods capable of limb regeneration has intrigued researchers for decades. Recent fossil evidence suggests an ancient origin of limb regeneration in tetrapods, as regeneration pathologies typically found among modern salamanders such as duplication or bifurcation of metacarpals, metatarsals and phalanges, as well as developmental asymmetry between the limbs within an individual, were reported in 300 million-year-old temnospondyl[1] and lepospondyl amphibians[2], ~80 million years before the estimated origin of stem salamanders. Recently, however, the notion of an ancient limb regeneration programme has been challenged by reports of salamander lineage-specific genes (LSGs) upregulated during regeneration[3–6]. One salamander LSG in particular, the *Prod1* gene, was shown to be required for proximodistal patterning during limb regeneration[7] and for ulna, radius and digit formation during forelimb development[8]. The existence of urodele LSGs expressed and involved in regeneration has lent support to the hypothesis that limb regeneration is a derived urodele feature[5,6]. Nevertheless, it remains unclear whether urodele LSGs are causally linked to the origin of limb regeneration or were integrated into a pre-existing regenerative programme. Appendage regeneration is also observed in living sarcopterygian (lobe-finned) fish such as the African lungfish *Protopterus*[9] and could have an even deeper origin, since basal actinopterygian (ray-finned) fish of the genus *Polypterus* can fully regenerate paired appendages, including the endochondral skeleton[10] (Fig. 1a). Nevertheless, the molecular bases of *Polypterus* and lungfish fin regeneration remains unexplored. Lungfishes, as the sister group to tetrapods[11,12], constitute the ideal model organisms to study the origin of limb regeneration in tetrapods. Nevertheless, limited taxonomic representation and scarce genetic resources have prevented in-depth comparisons of lungfish and salamander regeneration programs.

To address this question, we have examined fin regeneration in lungfishes, focusing on the morphological and molecular mechanisms leading to blastema formation. We find that, at a morphological and cellular level, lungfish fin regeneration is strikingly similar to salamander limb regeneration. To compare the genetic programs deployed during appendage regeneration in lungfish and salamanders, we produced a *de novo* assembly of the lungfish regenerating blastema, as well as additional transcriptomes of fin blastemas (FBs) and non-regenerating fins (NRFs). Our differential gene expression analysis reveals remarkable parallels between lungfish and salamander appendage regeneration, including strong downregulation of genes encoding muscle proteins, and conversely, upregulation of genes encoding matrix metalloproteinases, stem cell factors, and those involved in oncogenesis and developmental processes. Furthermore, we show that MARCKS-like protein (MLP), a molecule upregulated shortly after wound healing and involved in the initial steps of regeneration in salamanders, is also upregulated during early lungfish fin regeneration stages. Finally, we identify lungfish LSGs overexpressed during fin regeneration and show that, as in salamanders, LSG expression is not limited to regenerating tissues. Taken together, the shared features of lungfish and amphibian appendage regeneration point to a common evolutionary origin, with new genes integrated into pre-existing regeneration programs.

## Results

### Fin regeneration in the South American lungfish

To gain insight into the evolutionary origin of limb regeneration, we examined morphological and molecular events underlying fin regeneration in the South American lungfish, *Lepidosiren paradoxa*. The phylogenetic position of lungfishes as the closest living relatives of tetrapods[11,12] makes them the ideal taxon to address this question (Fig. 1a). The whip-like pectoral and pelvic fins of *Lepidosiren* lack pre- and post-axial radial elements and consist of a series of distinct cartilaginous elements, or mesomeres (Supplementary Fig. 1a,b). Among our wild-caught specimens, 7 out of 37 (18.9%) displayed potential regeneration pathologies consisting of bifurcations along the anteroposterior axis of the fin (Supplementary Fig. 1c,d), not unlike those observed in urodeles[13]. Furthermore, the percentage of pathological fins observed was similar to rates reported in regeneration studies on *Protopterus* under laboratory conditions (22%)[14]. These observations suggest that fin regeneration is a common occurrence in natural lungfish populations.

On monitoring pectoral fin regeneration after amputation, we found that a blastema formed during the first 3 weeks post-amputation (wpa), after which the regenerating fin continued to extend distally (Fig. 1b). At 1 wpa, the injured area was covered by a wound epidermis (WE) and bromodeoxyuridine (BrdU) labelling revealed very few proliferating cells (Fig. 1c,f). At 2 wpa, tissue disorganization and the loss of purple cartilage staining indicated loss of cell–cell contact and breakdown of extracellular matrix (ECM), consistent with histolysis (Fig. 1d). Still at 2 wpa, the WE thickened to form an apical ectodermal cap (AEC) and a blastema was formed immediately subjacent to the WE. Cell proliferation in the 2 wpa blastema occurred in epithelial cells, and in presumptive muscle cells flanking the cartilage skeleton (Fig. 1g). At 3 wpa, new cartilage condensation was apparent, an indication that cell differentiation and repatterning of the fin tissue was underway (Fig. 1e). Cell proliferation was detected in the blastema and in cells flanking the cartilage proximal and distal to the amputation site (Fig. 1h). At this stage, a basement membrane between the distal epithelium and the underlying blastemal cells was not visible (Supplementary Fig. 2). In salamanders, limb amputation triggers formation of the WE, histolysis, loss of a basement membrane underlying the AEC, dedifferentiation, subsequent blastema formation and repatterning[15], in a similar progression as described here for lungfish. These observations suggest that the cellular events involved in blastema formation in lungfishes and salamanders are equivalent.

### Differential gene expression in the lungfish fin blastema

Previous studies have examined the genetic pathways controlling limb regeneration using high-throughput DNA sequencing methods[16–25]. Here, we generated a *de novo* assembly of the 3 wpa lungfish FB reference transcriptome (Supplementary Table 1). To identify genes differentially expressed during regeneration, we produced additional transcriptomes of FBs at 3 wpa and NRF tissues in biological triplicates, with Pearson correlation coefficients among replicates >0.97 (Supplementary Tables 2 and 3). In total, we obtained read counts (Supplementary Data 1) and transcripts per million (TPM) values for 122,014 transcripts of our reference transcriptome (Supplementary Data 2). Our analysis of 28,844 transcripts annotated to human orthologs (Supplementary Data 3) revealed 12,810 unique genes, from which 4,415 showed significant *P* values among replicates (Student's *t*-test and false discovery rate *P* value correction, *P* < 0.05). Among these, there were 2,034 genes that showed greater than twofold expression change: 769 genes were downregulated and 1,265 genes were upregulated in FB relative to NRF (Supplementary Data 4).

We found many similarities between the expression profiles of our lungfish data set and amphibian blastemas at or around 2 wpa[17,19,23–25]. Among the downregulated genes, the majority

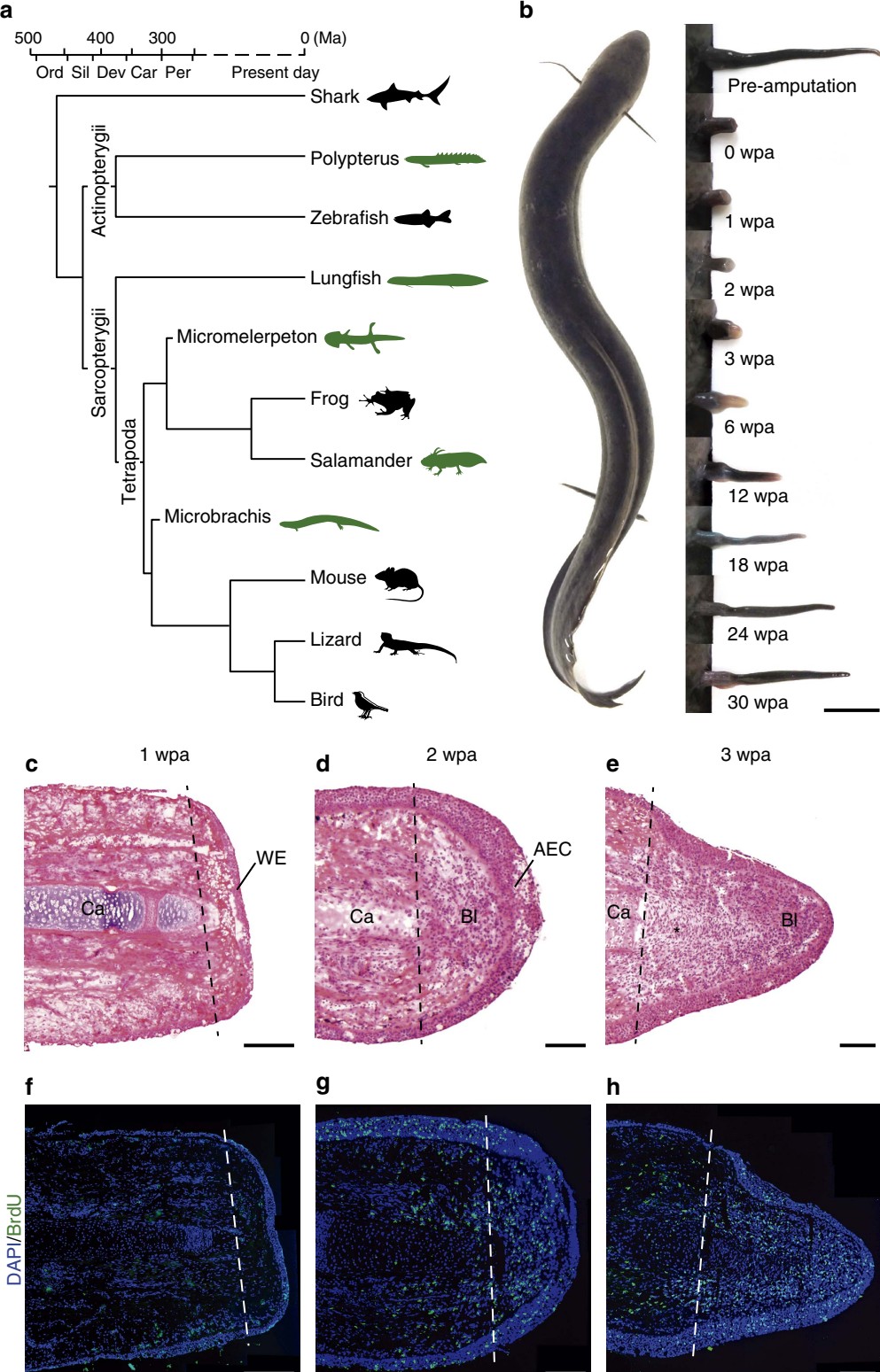

**Figure 1 | Fin regeneration and blastema formation in the *L. paradoxa*.** (**a**) Vertebrate phylogenetic tree, highlighting in green extant and extinct taxonomic groups capable of complete appendage regeneration among actinopterygians (*Polypterus*), sarcopterygian fish (lungfish) and tetrapods (salamanders, *Micromelerpeton* and *Microbrachis*). (**b**) Pectoral fin regeneration monitored for 30 wpa in an adult *L. paradoxa* specimen. (**c,f**) At 1 wpa, formation of a WE occurs with minimal mitosis. (**d,g**) At 2 wpa, AEC forms and cells accumulate distally and form a blastema, cell proliferation occurs in regions flanking the cartilage and in the blastema. (**e,h**) At 3 wpa, blastema extends distally, new cartilage is forming and cell proliferation intensifies. Haematoxylin and eosin stained sections are shown (**c–e**). Proliferating cells during fin regeneration are shown (green), the nuclei of all cells is stained (blue) (**f–h**). Dashed lines denote amputation planes. Asterisk denotes newly formed cartilage. Scale bars of 10 mm (**b**), 1 mm (**c,f**) and 0.5 mm (**d,e,g,h**). Ca, cartilage; Bl, blastema; Ma, million years ago.

of top 25 enriched gene ontology (GO) categories (Supplementary Data 5) were associated with muscle function (Supplementary Fig. 3a), a trend previously described for limb regeneration in both salamanders and *Xenopus* tadpoles. Similarly, analysis of Kyoto Encyclopedia of Genes and Genomes (KEGG) pathways (Supplementary Data 6) of FB downregulated genes showed that four of the five enriched terms were directly related to muscle function (Supplementary Fig. 4a). Conversely, upregulated GO categories were enriched for terms related to ECM, morphogenesis and cell cycle (Supplementary Fig. 3b), all consistent with cellular processes characteristic of limb regeneration. Accordingly, KEGG pathway enriched terms included cell cycle, ECM-receptor interaction, focal adhesion, p53 signalling and pathways in cancer (Supplementary Fig. 4b). Quantitative PCR (qPCR) profiles of 16 up or downregulated targets corroborated the transcriptome expression data (Supplementary Fig. 4c).

A closer inspection of the differentially expressed genes revealed further commonalities between lungfish and amphibian regeneration (Fig. 2). Several genes downregulated in axolotl blastemas were also downregulated in lungfish blastemas, such as those encoding proteins under the functional categories of muscle (*Des, Tpm1, Acta2, Actc1, Myl2, Ttn, Tnnc1, Smpx, Myoz1, Tnnt3, Mybpc2* and *Ckm*), tight junction (*Cldn10, Actn2* and *Actn3*) and calcium homeostasis (*Ryr1, Casq1, Atp2a1* and *Sln*). These expression changes are consistent with extensive histolysis and muscle degeneration, which occur during wound healing and are sustained until blastema formation and growth. Among upregulated genes, the lungfish and salamander blastema gene sets were highly congruent[17,19,23,24]. Our results showed that genes encoding cell cycle components including *Mcm2, Pcna, Cdc20, Plk1* and *Ccnb1*, focal adhesion genes such as *Itga8, Itgb5* and *Itga11*, as well key genes encoding ECM components and modulators *Mmp8, Mmp9, Mmp13, Timp1, Col5a1, Col12a1, Emilin1* and *Fn1*, all typically overexpressed in salamander blastemas, were also upregulated in the lungfish blastema.

Furthermore, as in urodeles, we observed overexpression of genes associated with cellular reprograming, such as the stem cell genes *c-Myc, Sall4, Sox2, Jarid2, Sall1, Zic2,* and limb development genes, including *Fgf1, Fgf10, Fgf16, Hoxd13, Hoxa13, Tbx5, Wnt5a, Wnt5b, Tgfb1* and *Tgfb2*[17,23]. Finally, the 51 differentially expressed lungfish genes listed above were compared with the expression profiles described in a recent comprehensive microarray study of axolotl limb regeneration[25]. Whereas 10 genes did not have corresponding probes, the correlation between gene expression fold change in our lungfish data set and the 2 wpa salamander blastema was ~65% (Supplementary Data 7).

**Upregulation of *Mlp* expression during early fin regeneration.** Recently, the extracellularly released factor MLP was shown to induce the initial proliferative response associated with appendage regeneration in axolotls[26]. Expression of the axolotl *Mlp* peaks during the first 24 h post-amputation, decreasing to basal levels after 4 days. Here we identified the lungfish MLP ortholog and showed that it contains the three conserved domains found in other vertebrate MLPs, including the effector domain and its key serines that are potential targets for phosphorylation by protein kinase C (PKC)[27] (Fig. 3a). Phylogenetic analysis showed that lungfish MLP groups with other vertebrate MLPs (Fig. 3b). As expected, in our 3 wpa FB, *Mlp* expression levels were similar to those seen in NRF (Supplementary Data 4). However, qPCR using two distinct *Mlp* primer sets revealed that, as seen in axolotls, lungfish *Mlp* is highly expressed at 1 day post-amputation (dpa), and expression levels gradually decrease at 7 and 14 dpa (Fig. 3c), only reaching basal levels at 3 wpa (Supplementary Data 4). This gradual decrease in *Mlp* expression is consistent with the slow regenerative process observed in lungfish fins. In sum, the robust early expression of *Mlp*, a key factor for initiation of limb regeneration in salamanders, also occurs during lungfish fin regeneration.

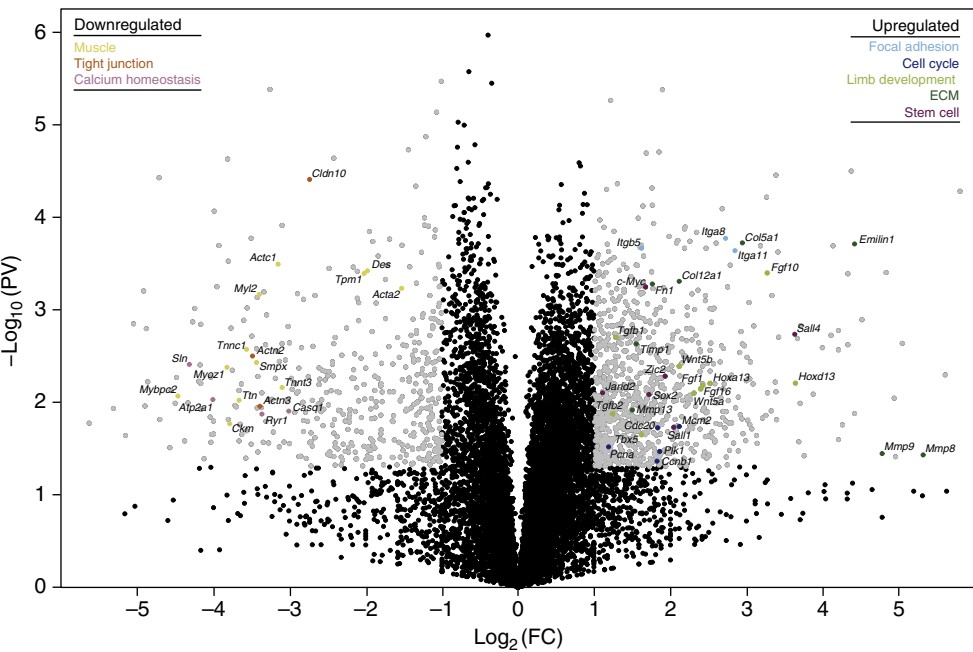

**Figure 2 | The genetic basis of lungfish fin regeneration.** Volcano plot showing differentially expressed genes between NRF tissue and 3 wpa FB. In a volcano plot, each dot represents a gene. Black dots are either below Student's *t*-test *P* value (<0.05) or fold change (>2) cutoffs. Grey dots are within established limits of *P* value and fold change. Key genes are shown colour coded according to eight categories: muscle, tight junction, calcium homeostasis, focal adhesion, cell cycle, limb development, ECM and stem cell. Fold change displayed in a log$_2$ scale and *P* values in $-\log_{10}$. PV, *P* value; FC, fold change.

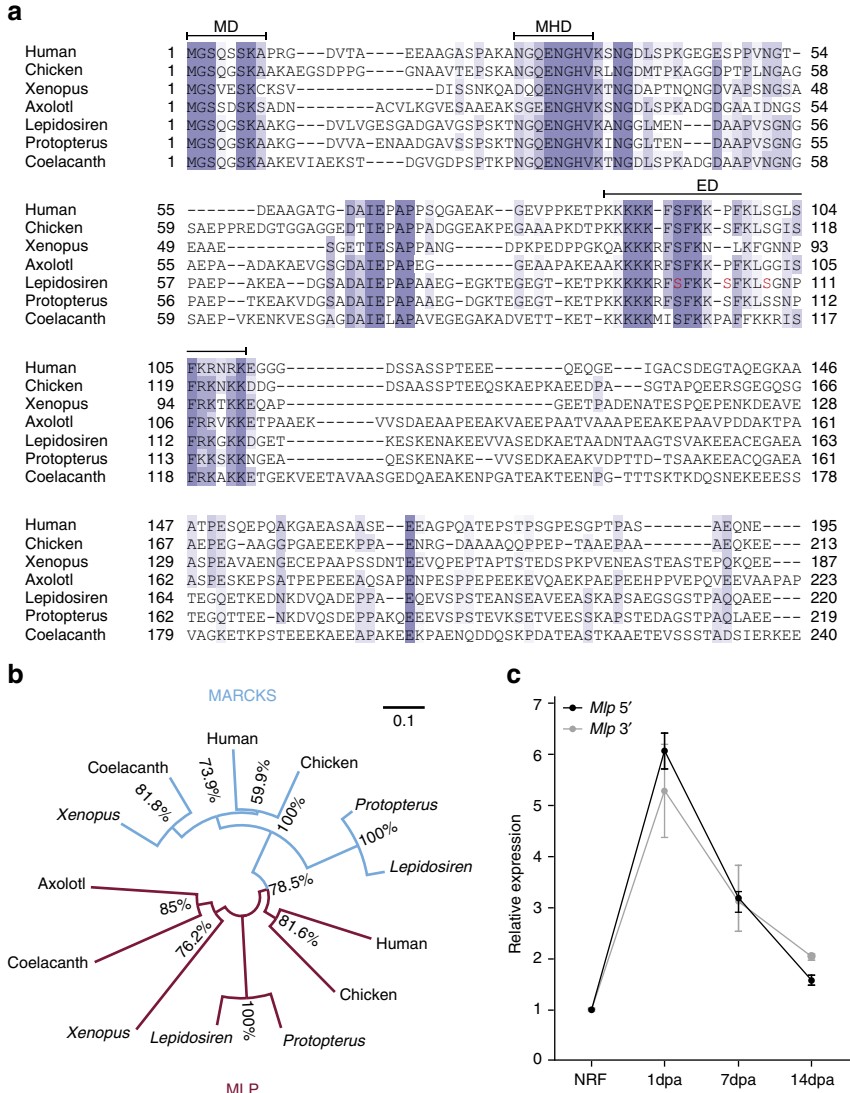

**Figure 3 | Lungfish *Mlp* expression on regenerating fin. (a)** Sequence alignment of MLPs as performed using a similarity matrix BLOSUM62 to compare sequences from human (*Homo sapiens*); chicken (*Gallus gallus*); frog (*Xenopus laevis*); axolotl (*Ambystoma mexicanum*); lungfishes *Lepidosiren paradoxa* and *Protopterus annectens*; and coelacanth (*Latimeria chalumnae*). Black bars highlight the three MLP conserved domains: myristoylated N terminus domain (MD), MARCKS homology domain (MHD) and effector domain (ED). Serines in red are potential target for phosphorylation by PKC. **(b)** Phylogenetic tree of MARCKS (blue) and MLP (red) protein orthologs was inferred using the Neighbour-Joining method. Bootstrap percentage values are shown next to the branches and the evolutionary distances were computed using the p-distance method. **(c)** Relative expression for the lungfish *Mlp*, using two sets of primers (*Mlp* 5′ in black, *Mlp* 3′ in grey), was obtained for FB, 1, 7 and 14 dpa, normalized to expression level in NRF to a relative value of 1.0 ($n = 1$ biological sample, three technical replicates); mean ± s.d.

**Lungfish-specific genes expressed in the fin blastema**. In salamander, LSGs are overexpressed during regeneration[4,6] and at least in one instance (*Prod1*), acquired an essential role[7,8]. Here we performed a pilot survey of lungfish putative LSGs associated with fin regeneration. We searched in our data set for transcripts with no annotation in the orthoMCL or Metazoan UniProt databases (Supplementary Data 8) and found 206 transcripts with complete open reading frames of more than 500 bp, and with at least 10 mapped reads in each of the 6 transcriptome replicates (Supplementary Data 9). From this list we retrieved 35 transcripts, with 19 transcripts significantly up or downregulated during fin regeneration (Supplementary Data 10 and Fig. 4a). Based on the presence of orthologs in the African lungfish *Protopterus annectens*[11] (Supplementary Fig. 5), we selected four LSGs for further analysis: two significantly upregulated and two that fell

below our statistical threshold (Supplementary Data 10 and Fig. 4a). The putative open reading frames of the four LSGs ranged from 801 to 2,376 bp, displayed no hits on the NCBI database or Sal-Site expressed sequence tag (EST) and gene expression databases (www.ambystoma.org), and encoded no known domains in c28232 and c29579 (Supplementary Fig. 5a,b), except for signal peptide sequences in LSGs c19141 and c19958 (Supplementary Fig. 5c,d). The LSGs are corroborated by extensive read coverage (Fig. 4b) and qPCR showed that candidate LSGs are expressed in multiple tissues in addition to the FB (Fig. 4c). We were able to confirm the four LSGs sequences by reverse transcription PCR followed by sequencing, and the bands obtained corresponded to the expected amplicon sizes (Supplementary Fig. 5e). Our findings suggest that these genes may have roles beyond those potentially associated with fin regeneration.

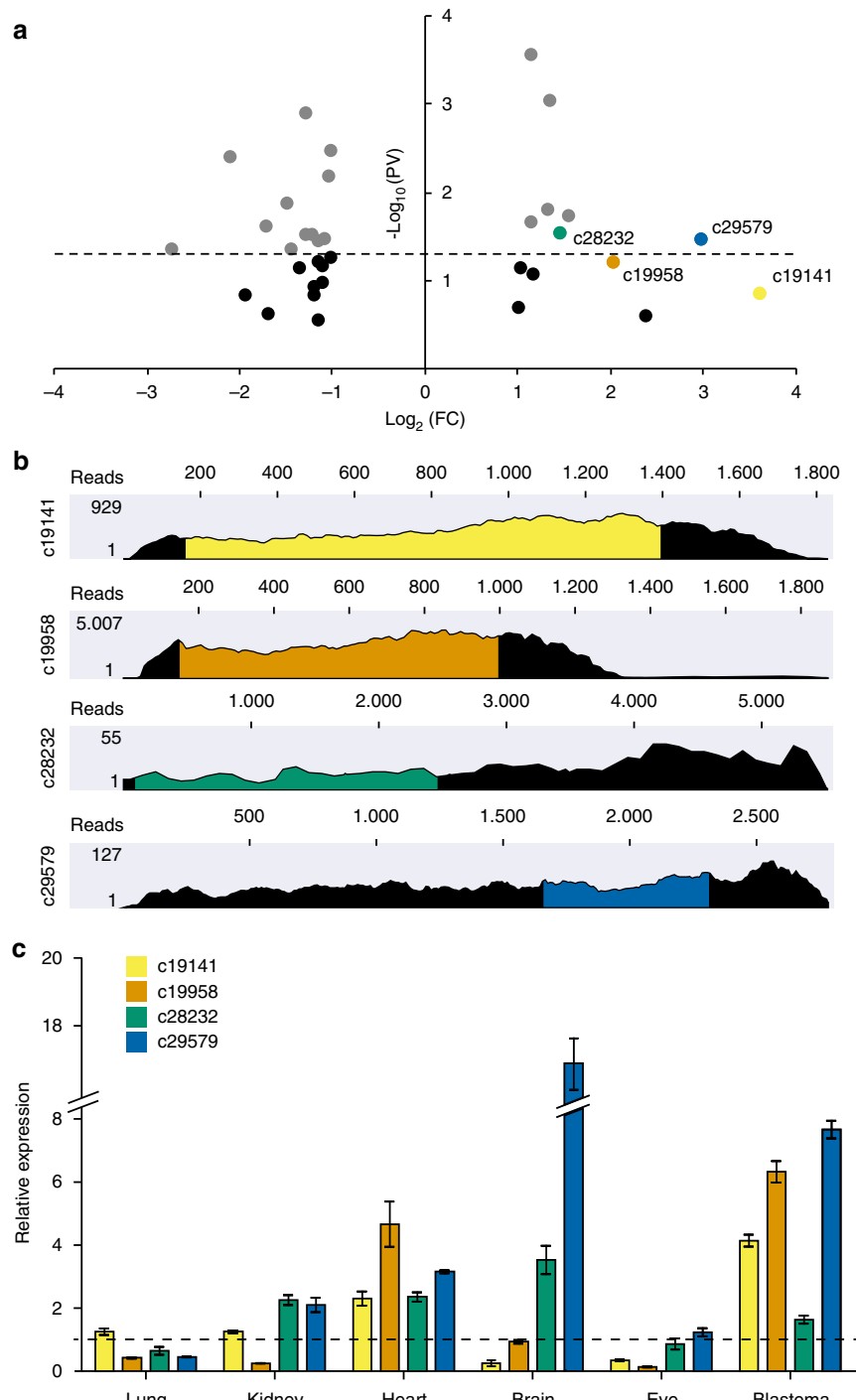

**Figure 4 | Lungfish LSG expression in the FB and other adult tissues. (a)** Volcano plot showing differentially expressed LSGs between NRF tissue and FB at 3 wpa. Each dot represents a gene and coloured dots denote four LSGs selected for further analysis (c19141, yellow; c19958, orange; c28232, green; c29579, blue). Black and grey dots represent genes below or within the $P$ value cut-off, respectively, which is denoted by a dashed line Student's $t$-test $P$ value ($<0.05$). Fold change is displayed in a $\log_2$ scale and $P$ value in $-\log_{10}$. **(b)** RNA-seq read coverage of the four candidate LSG transcripts shown in black, the predicted open reading frames (ORF) is shown in colour for each LSG, according to colour scheme in **a**. **(c)** Relative expression for the four, colour coded, candidate LSGs in various tissues is shown, normalized to expression level in NRF to a relative value of 1.0 (dashed line) ($n = 1$ biological sample, three technical replicates); mean $\pm$ s.d.; PV, $P$ value; FC, fold change.

## Discussion

Here we showed that *Lepidosiren* collected from natural sources display a high percentage of pathological fins, similar to regeneration pathologies found in natural salamander populations[28,29]. Furthermore, the morphological steps leading to blastema formation in the lungfish and salamanders are strikingly similar and involve extensive histolysis, AEC formation and a blastema that lacks a basement membrane. The source of blastema cells, whether chiefly derived from dedifferentiation (adult mode) or from resident stem cells (larval mode), as recently demonstrated in newts[30], remains unclear.

*De novo* assembly of the lungfish transcriptome and differential gene expression analysis revealed notable parallels between lungfish and salamander appendage regeneration[17,19,23,24], including strong downregulation of muscle proteins, accompanied by upregulation of matrix metalloproteinases, stem cell and developmental genes. In urodele blastemas, *Tgfb1* and its target fibronectin (*Fn1*) are thought to provide directional guidance to blastema cells[31]. Likewise, we observed upregulation of *Tgfb1* and *Fn1* in lungfish blastemas. In addition, we showed that *Mlp*, a gene upregulated during early stages of limb regeneration and recently proposed as a regeneration-initiating molecule in salamanders[26], is also upregulated during the initial stages of lungfish fin regeneration. Altogether, our findings strongly suggest that the morphological parallels between lungfish and axolotl regeneration result from equivalent gene expression profiles during regeneration.

In one FB and two NRF transcriptome replicas, no reads were mapped to *Ffg8*, preventing differential gene expression assessment (see Supplementary Data 4). *Fgf8* is a typical apical ectodermal ridge (AER) marker in the tetrapod limb, yet in zebrafish it is expressed later in the apical fold (after 36 hpf), whereas other *Fgfs* such as *Fgf16* and *Fgf24* are expressed earlier in the AER (before 34 hpf)[32–34]. While an *Fgf24* gene ortholog was not found in our lungfish reference transcriptome, *Fgf16* was upregulated in the lungfish blastema. Therefore, it is possible that FGF expression in the 3 wpa lungfish blastema is more compatible to the early fin bud of teleosts.

Our pilot search for lungfish LSGs identified at least four genes that, as in salamanders, are upregulated during fin regeneration. In recent years, a large number of LSGs were found in various organisms and in many cases shown to be integrated into existing gene networks, including those involved in development[35]. A study in *Drosophila* identified 195 LSGs and showed that ∼30% of these new genes resulted in lethal phenotypes when silenced by RNA interference[36]. Urodele LSGs, like those of lungfish, also show expression in various organs other than regenerating tissues[6], indicating that their roles are not limited to regenerating tissues. Within this framework, we propose that salamander and lungfish LSGs may not be necessarily linked to the origin of regeneration capacities. Alternatively, LSGs may have been subsequently integrated into a pre-existing appendage regeneration programme.

Complete fin regeneration observed in *Polypterus* suggests that appendage regeneration may have evolved at the base of all bony fish[10], nevertheless, little is known of the molecular mechanisms underlying *Polypterus* fin regeneration. Limited appendage regeneration capacities are also observed in teleost fish and mammals, which can regenerate fin rays and digit tips, respectively, but not the endochondral elements[31]. Future studies comparing the molecular programs deployed during regeneration across various taxa may help determine how regeneration programs evolved in lineage specific ways to accomplish the similar functional outcomes.

It has been over 40 years since fin regeneration in lungfishes was documented and its significance in the context of tetrapod limb regeneration has remained underappreciated. Similar to urodeles, lungfishes can also fully regenerate tails[14], suggesting that this and other remarkable salamander-like regenerative capacities could have an ancient evolutionary origin. Our study shows deep morphological and molecular similarities between urodele and lungfish regeneration, establishing the lungfish as a valuable model for future studies on the evolution of regeneration in vertebrates.

## Methods

**Animals and surgical procedures.** This study was approved by IBAMA/SISBIO under license number 47206–1. All experimental procedures and animal care were conducted in accordance to the Ethics Committee for Animal Research at the Universidade Federal do Pará, under the approved protocol number 037-2015. A total of 37 adult *Lepidosiren paradoxa* specimens, ranging from 64 to 95 cm in length and 0.9 to 3.71 kg in weight, were obtained from natural sources in the state of Pará, Brazil. No statistical methods were used to predetermine sample size. *In vivo L. paradoxa* experiments were not randomized and no blind tests were applied. Animals were kept in individual tanks with regular water changes and fed once a day. For fin regeneration studies, animals were anaesthetized in 0.1% clove oil diluted in system water. Pectoral fins were bilaterally amputated at ∼1 cm distance from the body. FBs were sampled at 1, 2 or 3 wpa and stored in RNAlater (Sigma-Aldrich) for RNA extraction or embedded in Tissue Tek O.C.T compound (Sakura Finetek) in dry ice, and then maintained in −80 °C freezer for cryosectioning.

**External morphology and histology of fin regeneration.** Pectoral fins were photographed weekly to document the changes on external morphology during regeneration. Photos were taken during a period of 30 weeks. For histology, frozen tissues were allowed to equilibrate to cryostat temperature (−20 °C) for 30 min and then 20 μm longitudinal sections were obtained on ColorFrost Plus microscope slides (Thermo Fisher Scientific). Sections were fixed in 3% paraformaldehyde for 5 min, rinsed twice in 0.01 M PBS, and dehydrated in graded ethanol series (70, 95 and 100%), for 2 min each. After drying at room temperature, slides were stored in −80 °C ultrafreezer. Sections stained with haematoxylin (Sigma-Aldrich) and eosin (Sigma-Aldrich) were imaged on a SMZ1000 stereoscope (Nikon). Pectoral fin was cleared and stained following standard protocol with modifications[37]. The fixation was performed using formaldehyde solution (10%) for 24 h. Samples were dehydrated using serial dilutions of ethanol (50, 70 and 95%) for 12 h each and absolute alcohol for 12 h (2 times). Cartilage staining was performed using 30 mg of Alcian blue in 100 ml of a 40% solution of glacial acetic acid and 60% absolute ethanol for 12–24 h, and a saturated solution of borax was used for neutralization for 48 h. Subsequent bleaching was performed using 10% solution of $H_2O_2$ in 0.5% KOH solution for 12 h.

**Cell proliferation.** For cell proliferation studies, BrdU was injected intraperitoneally into anesthetized lungfish (80 mg kg$^{-1}$ of body weight), 24 h before tissue collection. For BrdU immunolocalization, sections were permeabilized in 1N and 2N HCl solution, followed by 0.1 M borate buffer and in PBS tween (0.1% tween in 0.01M PBS). Unspecific labelling was blocked with 5% normal goat serum diluted in 0.01 M PBS with 0.3% TritonX-100 for 1 h at room temperature. Sections were then incubated with mouse anti-BrdU primary antibody (1:200, Sigma-Aldrich, cat. number B8434) in 0.01 M PBS with 1% bovine serum albumin and 0.3% TritonX-100 overnight at 4 °C. On the next day, sections were incubated with the fluorochrome-conjugated secondary antibody (1:400, Sigma-Aldrich, cat. number SAB4600238) for 2 h at room temperature and slides were mounted and counterstained with Fluoroshield with DAPI (Sigma-Aldrich).

**Library preparation and Illumina sequencing.** Total RNA extraction from FB or NRF tissue for transcriptome or qPCR was achieved using TRIzol Reagent (Life Technologies) according to the manufacturer's protocol. RNA samples were further purified using RNeasy Mini Kit (Quiagen) and treated with DNaseI (Quiagen), according to the manufacturer's protocol. Total RNA (0.3 or 0.5 μg) was reverse transcribed using the SuperScript III First-Strand Synthesis SuperMix (Life Technologies). To establish a lungfish blastema reference transcriptome, a HiSeq Illumina run was obtained from a blastema library of 100 bp paired-end reads (SRX1411314). Transcript abundance was estimated based on six independent Illumina 50 bp single-end runs, three runs from FB libraries (SRX1411321, SRX1411322 and SRX1411324) and three runs from NRF libraries (SRX1411325, SRX1411326 and SRX1411327).

**Bioinformatic analysis.** *De novo* assembly of lungfish transcriptome was performed using Trinity with default parameters[38] and the data set was registered under Bioproject PRJNA301439 in NCBI database. The functional annotation was performed with BLASTx comparison against Metazoan data set of UniProt proteins and retrieving GO terms using Blast2GO (ref. 39). CLC genomic workbench was used to map reads on transcripts with default parameters (CLC bio, Aarhus, Denmark).

For further lungfish transcriptome characterization, transcripts were clustered by human orthologs using BLASTx against Human NCBI Refseq database (11/2014), with an e-value of 10$^{-10}$, as described for a previous axolotl transcriptome[10]. For each human homologue gene cluster (HHGC), expression was calculated in TPM for the six conditions based on the read count sum of lungfish transcripts included in the cluster. The differential gene expression test was performed using *t*-test of CLC genomic workbench software, considering two conditions (NRF and FB) and three independent biological replicates. The HHGCs were analysed in GO terms and KEGG pathway enrichment using web-based tool DAVID[40]. GO terms and KEGG pathways were first selected based on a Bonferroni corrected *P* value cut-off of 0.01 and then ranked according to enrichment score. GO word clouds were generated using the online tool Wordle (www.wordle.net). The comparison between differentially expressed lungfish HHGCs and axolotl regenerating limb microarray probes[25] was based probe annotation. A linear

regression was used to determine the correlation between the fold changes of microarray probes (t0 versus t14) and lungfish RNA-seq (NRF versus FB), in log2 scale. The four lungfish LSGs identified, c19141_g1_i1, c19958_g1_i1, c28232_g1_i1 and c29579_g2_i1, correspond to transcript isoforms of the contigs c19141, c19958, c28232 and c29579, respectively. In the main text and figures, for simplicity, LSGs were referred to only by their contig identification numbers.

**Reverse transcription-PCR verification of lungfish LSGs.** Total RNA was isolated from heart tissue using TRIzol Reagent (Life Technologies) according to the manufacturer's protocol. Total RNA (0.3 μg) was reverse transcribed using the SuperScript III First-Strand Synthesis SuperMix (Life Technologies). The four lungfish LSGs selected for further analysis were PCR amplified (Life Technologies) in final volume of 50 μl under the following cycle conditions: 10 min at 95 °C followed by 35 cycles of 30 s at 95 °C, 30 min at 58 °C, and 3 min 72 °C. PCR products were cloned into PCR4-TOPO TA cloning vector (Life Technologies) and verified by direct sequencing. The following oligonucleotide sequences were used: c28232-F: 5′-TATGCAGAATGGCATCAGACA-3′; c28232-R: 5′-TCACCCTCC ACATAATTCACC-3′; c19958-F: 5′-TTTTGCTGGCAGTCAGTGTC-3′; c19958-R: 5′-TGTGTCCCAGCCACACTATT-3′; c29579-F: 5′-GGAAGTCCCCA AAAGGATACA-3′; c29579-R: 5′-CACCCTGTTAGCTGTTGCATT-3′ c19141-F: 5′-CCCCAGATCATACCAGGAAAT-3′; c19141-R: 5′-CTTGTGCCAACCCTA AGGAAT-3′.

**Phylogenetic analysis.** Sequence alignments were performed using ClustalW and edited in Jalview 2.9.0b2, using similarity matrix BLOSUM62 score. The evolutionary history was inferred using the Neighbour-Joining method[41]. The percentage of replicate trees in which the associated taxa clustered together in the bootstrap test (1,000 replicates) are shown next to the branches[42]. The evolutionary distances were computed using the p-distance method[43] and are in the units of the number of amino acid differences per site. The analysis involved 13 amino acid sequences and all positions containing gaps and missing data were eliminated, resulting in a total of 135 positions in the final data set. Evolutionary analyses were conducted in MEGA7 (ref. 44). The accession numbers used for alignments and phylogeny of MARCKS and MLP proteins, respectively, were as follows: *Homo sapiens,* (NP_002347.5 and AAH66915.1), *Gallus gallus,* (NP_990811.1 and NP_001074187.1), *Xenopus laevis* (NP_001080075.1 and NP_001108274.1), *Ambystoma mexicanum* (AMO27486.1), *Latimeria chalumnae* (XP_006004411 and XP_005995649.1). *L. paradoxa* sequences were obtained from the current study and *P. annectens* sequences from publically available RNA-seq run[17].

**qPCR.** qPCR of NRF and FB tissue was performed on biological replicates in triplicate (and triplicate technical qPCR replicates). Lungfish LSG analysis was performed on triplicate technical replicates and utilizing different tissues (lung, kidney, heart, brain, eye, NRF and FB). LSG expression in NRF was used as a reference to obtain relative expression levels the other tissues assayed. *Mlp* relative expression levels were assessed in triplicate technical replicates at 1, 7 and 14 dpa, and compared to expression levels in NRF. All experiments were performed using SYBR Green PCR Master Mix (Applied Biosystems) in final volume of 10 μl. Lungfish gene specific oligos for qPCR assays were designed using Primer Express Software Version 3.0 (Applied Biosystems) and used in final concentration of 2 mM to each primer. qPCR reactions were performed in the 7,500 real-time PCR System (Applied Biosystems) under the following cycle conditions: 2 min at 50 °C, 10 min at 95 °C followed by 40 cycles of 15 s at 95 °C, 1 min at 60 °C. Relative messenger RNA expressions were calculated using the $2^{-\Delta\Delta CT}$ method[45]. Oligonucleotides used are listed in Supplementary Data 11. The ΔCTs were obtained from CT normalized with POLR1C levels in each sample (Supplementary Data 12).

**Statistical analysis.** Digital gene expression was based on *t*-test with mean TPM value between NRF and FB conditions, for each transcript and HHGC. A transcript or HHGC is considered as differentially expressed if its fold change is superior to 2 or inferior to −2 and *P* value is inferior to 5%. GO and KEGG pathway enrichment analyses were performed using DAVID, based on a modified Fishers Exact *P* value, the EASE Score Threshold[33]. qPCR analysis data were analysed by Student's *t*-test (*P* value < 0.05), parametric, two-tailed test and was performed using GraphPad Prism version 5.0 for Windows (GraphPad Software, San Diego California USA, www.graphpad.com.).

**Data availability.** Sequence data that support the findings of this study have been deposited in GenBank with the following BioProject accession numbers: PRJNA301439, three from FB libraries (SRX1411321, SRX1411322 and SRX1411324) and three from NRF libraries (SRX1411325, SRX1411326 and SRX1411327). The four lungfish LSG sequences obtained from cDNA have been deposited in GenBank under the following accession numbers: KX534208 (c19141), KX534209 (c29579), KX534210 (c19958) and KX534211 (c28232). The authors declare that all other relevant data supporting the findings of this study are available on request.

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

## Acknowledgements

We thank Marcelo Pinheiro for assistance in collecting lungfishes. We also thank Guilherme Dutra and Patricia Schneider for critical reading of the manuscript. This work was funded by CAPES/CSF/PVE Program Grant 2813/2014 and CAPES/Alexander von Humboldt Foundation fellowship (to I.S.), postdoctoral fellowship from CAPES to C.M.C.

## Author contributions

A.F.N., C.M.C., J.L., R.N.M., S.D. and I.S. designed the research; A.F.N., R.N.M. and J.L. performed histological analyses. C.M.C., G.N.F-L., I.S. and J.L. performed and analysed qPCR data. C.F. constructed libraries, produced and analysed transcriptome data. M.R., C.T.A., S.D., C.M.C., J.L. and I.S. analysed transcriptome data; A.F.N., C.T.A., S.D. and I.S. wrote the manuscript with input from all authors. I.S. supervised this work.

## Additional information

**Competing financial interests:** The authors declare no competing financial interests.

**Publisher's note**: 

