## [Peer Review File · Nature Communications]

Reviewers' comments:

Reviewer #1 (Remarks to the Author):

The MS by Schiender et al. presents interesting and novel data concerning similarities between lungfish fin and urodele limb regeneration. They show, as much as can be shown with limited but strategic tissue sampling, that lungfish fin and urodele limb regeneration share anatomical, histological, and gene expression similarities. The transcriptomic approaches are typical and valid and the transcriptome data are high quality. Inclusion of three replicate biological samples for non-regenerating and regenerating tissues meets minimal standards and is sufficient in my mind, especially since the authors verify results with qPCR and also compare results to other studies, although I think an important transcriptome study was overlooked on salamander limb regeneration that could provide additional insight and make the manuscript stronger. The writing is clear and the paper is likely to appeal to a wide readership. Below are some comments meant to improve the manuscript.

Line 80: The idea that MLP triggers regeneration is a bit overstated. There are maybe tens to hundreds of genes whose knockout would phenocopy what was observed for MLP. I would reword.

Line 96: Could modify this sentence to include a reference for urodeles. "the anteroposterior axis of the fin (Supplementary Fig. 1c, d) not unlike those observed in urodeles",..... reference is Thompson et al. 2014. *Regeneration* 1:27-32.

Lines 118-119: The most comprehensive analysis of urodele limb regeneration should be referenced here....reference is Voss et al. 2015. *Regeneration* 2:120-136. The gene-by-gene expression profiles from this study are easily searched here:
<http://www.ambystoma.org/genome-resources/20-gene-expression>

Line 135: The observation that muscle genes are down-regulated relative to controls is discussed in some detail in Voss et al 2015 and probably should be referenced. Care should be taken to indicate more clearly the direction of expression change between FB and NRF in the manuscript and supplemental files.

Line 175: Here is the expression profile of Mlp in axolotl from Sal-Site. The expression does increase significantly by 12 hrs and does show the gradual decrease as described, but initial basal expression is also high and the magnitude of increase is much lower than is observed in lungfish. Is this gene more highly expressed initially in lungfish than axolotl?

http://www.ambystoma.org/index.php?option=com_content&view=article&id=109&Itemid=2&qtype=2&pid=axo29615

Lines 182-188: It would be informative to see if the putative lungfish LSGs are not represented in Sal-Site EST and gene expression databases, which are enriched for urodele regeneration genes.

Lines 215-222: I have not seen it reported that there is strong transcriptional up-regulation of shh in studies of limb regeneration. Also, EVI5 is not strongly transcriptionally up-regulated in axolotl, although it is weakly upregulated at the 2nd punctuated step of transcriptional regulation in the axolotl which is associated with cell cycle ontologies...see Voss et al 2015. http://www.ambystoma.org/index.php?option=com_content&view=article&id=109&Itemid=2&qtype=2&pid=axo11271

The #19 reference cited is for a proteomic study that may have no bearing on this. I would delete this paragraph or maybe focus on something different, like FGF pathway...e.g. why wasn't FGF8 identified as enriched?

Lines 223-228: While it is timely to focus on mlp, almost certainly there are more parallels than simply the up-regulation of mlp between lungfish and axolotl. Can't more be done here. I am just concerned about single gene / single paper focus when clearly there is much more going on during the initial transcriptional response. The field is often a bit too myopic in wanting regeneration to depend upon a silver bullet.

Lines 247-248: It's not simply a question of how many times appendage regeneration evolved, it is also how regeneration programs change in lineage specific ways to accomplish the same functional outcomes.

Reviewer #2 (Remarks to the Author):

The manuscript "Tetrapod limb regeneration is an ancient sarcopterygian feature" by Nogueira et al. presents novel data on lungfish fin regeneration. For this, the authors draw upon two lines of evidence: 1. a morphological study detailing the steps from amputation, over wound healing, blastema initiation and outgrowth by application of classic histological methods; and 2. by investigating the genetic basis of lungfish regeneration using a de novo transcriptome assembly of a 3 week blastema as well as differential gene expression analysis.

This integrated approach revealed that the morphological steps involved in lungfish fin regeneration are remarkably similar to those observed in salamanders, the only extant tetrapods capable of full limb regeneration, including distinct features such as e.g. the loss of a basement membrane. Moreover, the authors demonstrate similar expression profiles of genes involved in amphibian and lungfish appendage regeneration, with patterns of down- and upregulation of gene sets highly comparable between salamanders and lungfish. Finally, they present data on lungfish LSGs playing an important role in the regeneration of fins albeit also expressed in other tissues not related to fin regeneration, a pattern that has previously been observed with respect to salamander LSGs.

These data suggest that lungfish and salamander regeneration are based on a common and ancient regeneration program in which in the course of sarcopterygian evolution, lineage specific genes have been integrated and in part have taken over integral roles in regeneration and as well as in other processes.

The paper presents highly relevant novel data on appendage regeneration that significantly

broadens the scope - both in terms of a phylogenetic perspective and the general patterns involved in vertebrate appendage regeneration. The presented data and the new insights they provide will undoubtedly be of great interest for a large audience and will have a strong impact on the field of regeneration research. Though it has long been known that lungfish have regenerative capacities similar to salamanders, the difficulty of retrieving and housing lungfish for scientific studies have prevented a more detailed understanding of the potential similarities and differences between lungfish and salamanders. It is impressive how the authors have overcome these difficulties and close a significant gap in our understanding of appendage regeneration with the presented data.

The experiments were thoroughly designed and executed and provide an appropriate overview over the morphological and genetic basis of lungfish fin regeneration. The similarities between both, the histological progression and the genetic profile of regeneration in salamanders are truly striking and indeed suggest a common regeneration programme in sarcopterygians or even all osteichthyans. The relevance of this observation for regenerative medicine is obvious and will undoubtedly influence future research approaches in regeneration research.

I have only a few minor comments, which are mostly of editorial nature listed below.

Beyond those, I strongly support publication of this manuscript in Nature Communications.

1. The high percentage of pathological fins is striking and yet another similarity to natural salamander populations lending another bit of support for the common program for appendage regeneration. I realize that reference numbers are limited, but the authors may want to include this aspect in an additional sentence or two (see e.g. Dinsmore CE, Hanken J. 1986 Native variant limb skeletal patterns in the red-backed salamander, *Plethodon cinereus*, are not regenerated. *J. Morphol.* 190, 191-200. (doi:10.1002/jmor.1051900204)
Dearlove GE, Dresden MH. 1976 Regenerative abnormalities in *Notophthalmus viridescens* induced by repeated amputations. *J. Exp. Zool.* 196, 251-261. (doi:10.1002/jez.1401960212)

2.Lines 45-50:: Note that the duplications and bifurcations are observed in the metacarpals/metatarsals and phalanges rather than the carpals and tarsals in the temnospondyl *Micromelerpeton*. In the lepospondyl taxa evidence for limb regeneration is actually based on a developmental asymmetry between the limbs on the right and left sides within an individual rather than a pathological morphology. The latter do show salamander-like tail regeneration, too.

3. The phylogenetic tree in Fig. 1a only depicts extant forms. Given the conclusion of the paper that the regeneration program may be a deep homology of osteichthyans, I would find it useful to add at least a few fossil lineages here as well, possibly in a time calibrated phylogeny to illustrate the temporal and taxonomic depths.

Reviewer #3 (Remarks to the Author):

In this work the authors characterize fin regeneration in a lungfish. They demonstrate that fin regeneration proceeds with analogous steps compared to salamander limb regeneration. Transcriptome analyses identify several genes that are regulated both during fish fin and salamander limb regeneration. Of particular interest is the regulatory pattern of MLP, which displays similar temporal dynamics in both salamanders as well as lungfish. Further, the authors also identify putative species specific genes with potential role in regeneration .

This is an interesting paper, which overall is well written. However there are several issues that need to be addressed.

The title is a clear over statement. Although one possible interpretation of the data is that limb regeneration in tetrapods is not a taxon specific trait but rather inherited, the data by no means prove such a strong conclusion.

The authors claim that malformations in the wild caught specimen are regeneration pathologies. How do they know they are not developmental malformations?

Lines 128-129. The de novo assembly identified 4396 genes with significant fold change and the authors write that 443 were downregulated and 848 were upregulated. It is difficult to understand the maths here.

The use of GO terms used to classify regulated genes is not meaningful and very arbitrary. Also, they write that 4 out 5 enriched terms were directly related to muscle function. I noted that many of those were related to cardiac muscle. That can't be particularly relevant in the context.

What do the authors mean by congruent data sets comparing fish and salamanders? That is very imprecise. The authors should give real numbers on similar and different regulatory patterns. Also, the stage of limb regeneration in salamanders, from which the data set was taken, should be defined.

The authors should at least validate by PCR at least those 4 lineage specific genes that were selected for further analyses (line 191) to show that the novel domain composition is not merely an in silico artifact.

Response to reviewers

We thank the reviewers for comments and suggestions, which have substantially improved our manuscript. Please find below our point-by-point response.

Reviewer #1 (Remarks to the Author):

The MS by Schiender et al. presents interesting and novel data concerning similarities between lungfish fin and urodele limb regeneration. They show, as much as can be shown with limited but strategic tissue sampling, that lungfish fin and urodele limb regeneration share anatomical, histological, and gene expression similarities. The transcriptomic approaches are typical and valid and the transcriptome data are high quality. Inclusion of three replicate biological samples for non-regenerating and regenerating tissues meets minimal standards and is sufficient in my mind, especially since the authors verify results with qPCR and also compare results to other studies, although I think an important transcriptome study was overlooked on salamander limb regeneration that could provide additional insight and make the manuscript stronger. The writing is clear and the paper is likely to appeal to a wide readership. Below are some comments meant to improve the manuscript.

Line 80: The idea that MLP triggers regeneration is a bit overstated. There are maybe tens to hundreds of genes whose knockout would phenocopy what was observed for MLP. I would reword.

Response: Sentence was reworded accordingly.

Line 96: Could modify this sentence to include a reference for urodeles. "the anteroposterior axis of the fin (Supplementary Fig. 1c, d) not unlike those observed in urodeles",..... reference is Thompson et al. 2014. *Regeneration* 1:27-32.

Response: We thank the reviewer for this suggestion, sentence and reference were included

Lines 118-119: The most comprehensive analysis of urodele limb regeneration should be referenced here....reference is Voss et al. 2015. *Regeneration* 2:120-136. The gene-by-gene expression profiles from this study are easily searched here: <http://www.ambystoma.org/genome-resources/20-gene-expression>

Response: we thank the reviewer for noting this oversight. The reference has now been included. We have also included an additional supplementary data table and text on the Results section comparing our lungfish up and downregulated gene list to the microarray data available for the corresponding axolotl orthologs (Voss et. al., 2015).

Line 135: The observation that muscle genes are down-regulated relative to controls is discussed in some detail in Voss et al 2015 and probably should be referenced. Care should be taken to indicate more clearly the direction of

expression change between FB and NRF in the manuscript and supplemental files.

Response: the above reference was included and instances where expression changes between FB and NRF were ambiguous have been clarified.

Line 175: Here is the expression profile of Mlp in axolotl from Sal-Site. The expression does increase significantly by 12 hrs and does show the gradual decrease as described, but initial basal expression is also high and the magnitude of increase is much lower than is observed in lungfish. Is this gene more highly expressed initially in lungfish than axolotl?

http://www.ambystoma.org/index.php?option=com_content&view=article&id=109&Itemid=2&qtype=2&pid=axo29615

Response: Indeed, if we consider the mean number of transcripts per million (TPM) of Mlp in our lungfish RNAseq runs (3.22 in NRF; 3.90 in FB), the initial basal expression is relatively high in lungfishes as well (Supplementary Data 4). Our qPCR analysis of Mlp expression relative to NRF (increased between 5 and 6 fold) does show a higher magnitude increase when compared to the profile available at Sal-Site (Voss et. al., 2015, Regeneration). However, qPCR analysis of axolotl Mlp performed by Sugiura et. al., 2016, Nature (extended data fig. 5E) shows an increase of approximately 17 fold change from mature tissue, which is considerably higher than what we observed for lungfish. One possible explanation to the variability seen between studies (Voss, 2015, Sugiura 2016 and this study) could be sampling. According to Sugiura, 2015, MLP expression increases mainly in the wound epithelium, so depending on the amount of tissue collected at early regeneration stages, particularly with regards to tissue proximal to the amputation plane, the wound epithelium cells might be over or underrepresented.

Lines 182-188: It would be informative to see if the putative lungfish LSGs are not represented in Sal-Site EST and gene expression databases, which are enriched for urodele regeneration genes.

Response: We thank the reviewer for this suggestion. We have now checked on the Sal-Site EST and gene expression databases for A. mexicanum and A. t. tigrinum and did not find orthologs to the 4 lungfish LSGs examined on our study. We have added a paragraph relative to this search on the Results section.

Lines 215-222: I have not seen it reported that there is strong transcriptional up-regulation of shh in studies of limb regeneration. Also, EVI5 is not strongly transcriptionally up-regulated in axolotl, although it is weakly upregulated at the 2nd punctuated step of transcriptional regulation in the axolotl which is associated with cell cycle ontologies...see Voss et al 2015.

http://www.ambystoma.org/index.php?option=com_content&view=article&id=109&Itemid=2&qtype=2&pid=axo11271

The #19 reference cited is for a proteomic study that may have no bearing on this. I would delete this paragraph or maybe focus on something different, like FGF pathway...e.g. why wasn't FGF8 identified as enriched?

Response: we thank the reviewer for pointing out these inaccuracies. We have removed mention of Shh and EVI5. In addition, we have now added a paragraph to the Discussion section concerning fgfs and a possible explanation as to why Fgf8 was not overexpressed in our dataset.

Lines 223-228: While it is timely to focus on mlp, almost certainly there are more parallels than simply the up-regulation of mlp between lungfish and axolotl. Can't more be done here. I am just concerned about single gene / single paper focus when clearly there is much more going on during the initial transcriptional response. The field is often a bit too myopic in wanting regeneration to depend upon a silver bullet.

Response: We understand the reviewer's concern and have toned down instances along the text concerning Mlp and regeneration initiating. We chose to focus on Mlp because among the genes shown to be upregulated early in regeneration, this is the one gene where functional studies (gain and loss of function assays) provide experimental support for its role. While we agree that future studies might identify more genes causally related to the initiation of limb regeneration, we believe that a broad comparative survey during the initial transcriptional response in lungfish is beyond the scope of this manuscript.

Lines 247-248: It's not simply a question of how many times appendage regeneration evolved, it is also how regeneration programs change in lineage specific ways to accomplish the same functional outcomes.

Response: We agree with the reviewer and as a result have incorporated this idea in the Discussion section.

Reviewer #2 (Remarks to the Author):

The manuscript "Tetrapod limb regeneration is an ancient sarcopterygian feature" by Nogueira et al. presents novel data on lungfish fin regeneration. For this, the authors draw upon two lines of evidence: 1. a morphological study detailing the steps from amputation, over wound healing, blastema initiation and outgrowth by application of classic histological methods; and 2. by investigating the genetic basis of lungfish regeneration using a de novo transcriptome assembly of a 3 week blastema as well as differential gene expression analysis. This integrated approach revealed that the morphological steps involved in lungfish fin regeneration are remarkably similar to those observed in salamanders, the only extant tetrapods capable of full limb regeneration, including distinct features such as e.g. the loss of a basement membrane. Moreover, the authors demonstrate similar expression profiles of genes involved

in amphibian and lungfish appendage regeneration, with patterns of down- and upregulation of gene sets highly comparable between salamanders and lungfish. Finally, they present data on lungfish LSGs playing an important role in the regeneration of fins albeit also expressed in other tissues not related to fin regeneration, a pattern that has previously been observed with respect to salamander LSGs.

These data suggest that lungfish and salamander regeneration are based on a common and ancient regeneration program in which in the course of sarcopterygian evolution, lineage specific genes have been integrated and in part have taken over integral roles in regeneration and as well as in other processes. The paper presents highly relevant novel data on appendage regeneration that significantly broadens the scope - both in terms of a phylogenetic perspective and the general patterns involved in vertebrate appendage regeneration. The presented data and the new insights they provide will undoubtedly be of great interest for a large audience and will have a strong impact on the field of regeneration research. Though it has long been known that lungfish have regenerative capacities similar to salamanders, the difficulty of retrieving and housing lungfish for scientific studies have prevented a more detailed understanding of the potential similarities and differences between lungfish and salamanders. It is impressive how the authors have overcome these difficulties and close a significant gap in our understanding of appendage regeneration with the presented data.

The experiments were thoroughly designed and executed and provide an appropriate overview over the morphological and genetic basis of lungfish fin regeneration. The similarities between both, the histological progression and the genetic profile of regeneration in salamanders are truly striking and indeed suggest a common regeneration programme in sarcopterygians or even all osteichthyans. The relevance of this observation for regenerative medicine is obvious and will undoubtedly influence future research approaches in regeneration research.

I have only a few minor comments, which are mostly of editorial nature listed below. Beyond those, I strongly support publication of this manuscript in Nature Communications.

1. The high percentage of pathological fins is striking and yet another similarity to natural salamander populations lending another bit of support for the common program for appendage regeneration. I realize that reference numbers are limited, but the authors may want to include this aspect in an additional sentence or two (see e.g. Dinsmore CE, Hanken J. 1986 Native variant limb skeletal patterns in the red-backed salamander, *Plethodon cinereus*, are not regenerated. *J. Morphol.* 190, 191-200. (doi:10.1002/jmor.1051900204) Dearlove GE, Dresden MH. 1976 Regenerative abnormalities in *Notophthalmus viridescens* induced by repeated amputations. *J. Exp. Zool.* 196, 251-261. (doi:10.1002/jez.1401960212)

Response: We thank the reviewer for directing our attention to this similarity. We have incorporated this information and the references on the Introduction and Discussion.

2.Lines 45-50:: Note that the duplications and bifurcations are observed in the metacarpals/metatarsals and phalanges rather than the carpals and tarsals in the temnospondyl *Micromelerpeton*. In the lepospondyl taxa evidence for limb regeneration is actually based on a developmental asymmetry between the limbs on the right and left sides within an individual rather than a pathological morphology. The latter do show salamander-like tail regeneration, too.

Response: We thank the reviewer for noting this inaccuracy. Paragraph has been revised accordingly.

3. The phylogenetic tree in Fig. 1a only depicts extant forms. Given the conclusion of the paper that the regeneration program may be a deep homology of osteichthyans, I would find it useful to add at least a few fossil lineages here as well, possibly in a time calibrated phylogeny to illustrate the temporal and taxonomic depths.

Response: We thank the reviewer for this suggestion. We have now modified the figure 1a accordingly.

Reviewer #3 (Remarks to the Author):

In this work the authors characterize fin regeneration in a lungfish. They demonstrate that fin regeneration proceeds with analogous steps compared to salamander limb regeneration. Transcriptome analyses identify several genes that are regulated both during fish fin and salamander limb regeneration. Of particular interest is the regulatory pattern of MLP, which displays similar temporal dynamics in both salamanders as well as lungfish. Further, the authors also identify putative species specific genes with potential role in regeneration .

This is an interesting paper, which overall is well written. However there are several issues that need to be addressed.

The title is a clear over statement. Although one possible interpretation of the data is that limb regeneration in tetrapods is not a taxon specific trait but rather inherited, the data by no means prove such a strong conclusion.

Response: We understand the reviewer's point of view but we do not feel that the title is unwarranted. Our study provides multiple lines of evidence, including similarities of fin pathologies on natural populations, comparable histological events, and extensive molecular equivalence. We conclude that these similarities collectively are best explained by an ancient sarcopterygian origin for appendage regeneration. This is the central hypothesis being tested here and was not a point of contention for the other two reviewers.

The authors claim that malformations in the wild caught specimen are regeneration pathologies. How do they know they are not developmental malformations?

Response: This is a valid point. While we cannot be absolutely certain, we inferred that these were most likely a result of regeneration errors because such pathologies have been described in amputation experiments performed on African lungfishes and shown to occur at a 22% frequency, which is very close to what we have observed among wild caught *Lepidosiren* (18.9%) and comparable to that observed in natural salamander populations. This information and relevant references have now been added to the Results section. In addition, adult South American lungfish collected from natural sources often have shortened appendages with regenerating blastemas on its tips, suggesting that losing fins as a result of bites is common (personal observation). Given that lungfishes have long lifespans, we estimate that a wild-caught adult lungfish is unlikely to possess the fins it originally did as larvae.

Lines 128-129. The de novo assembly identified 4396 genes with significant fold change and the authors write that 443 were downregulated and 848 were upregulated. It is difficult to understand the maths here.

Response: We thank the reviewer for bringing this discrepancy to our attention. We have amended the text accordingly

The use of GO terms used to classify regulated genes is not meaningful and very arbitrary. Also, they write that 4 out of 5 enriched terms were directly related to muscle function. I noted that many of those were related to cardiac muscle. That can't be particularly relevant in the context.

Response: We understand the reviewer's concern. We nevertheless would like to point out that GO terms are a valuable tool to identify broad patterns and to indicate which gene assemblages are enriched or overrepresented. Many genes will fall into more than one GO category and a GO term such as cardiac muscle also contains general muscle genes; it doesn't mean that cardiac-restricted genes *per se* were overexpressed in our dataset. This overall approach is used widely for transcriptome data, and it is accepted that it merely provides broad trends in the gene expression dataset rather than an absolute metric.

What do the authors mean by congruent data sets comparing fish and salamanders? That is very imprecise. The authors should give real numbers on similar and different regulatory patterns.

Response: In the Results section, we highlight 51 genes that are commonly up or downregulated in lungfish and salamanders. Several of those genes have been experimentally linked to key steps in regeneration by previous papers. We have taken this approach because different data sets can vary significantly relative to how many genes were assayed and at what stage,

making it oftentimes difficult to directly compare RNAseq datasets and/or microarray data gene-by-gene. However, we have taken the reviewer's concern into account and we have now included an additional supplementary data table and text on the Results section comparing the 51 lungfish up and downregulated gene list to the microarray data available for the corresponding axolotl orthologs (Voss et. al., 2015).

Also, the stage of limb regeneration in salamanders, from which the data set was taken, should be defined.

Response: The stage of axolotl blastema being compared (2 wpa) was defined in line 132 (now line 140)

The authors should at least validate by PCR at least those 4 lineage specific genes that were selected for further analyses (line 191) to show that the novel domain composition is not merely an in silico artifact.

Response: We have now PCR-amplified from cDNA, cloned and sequenced-confirmed the 4 LSGs selected for further analyses. Their sequences were submitted to GenBank and validate the findings reported here. Accession numbers for these sequences have now been included in our methods. In addition, our supplementary Fig. 5 shows that the LSGs identified here for the South American lungfish are orthologous to sequences from an assembly from the African lungfish (Amemiya et al., 2013).

REVIEWERS' COMMENTS:

Reviewer #1 (Remarks to the Author):

The revised manuscript adequately addressed reviewer concerns, the manuscript is much improved.

Reviewer #2 (Remarks to the Author):

The reviewers have incorporated my comments to my full satisfaction. This is a very interesting study that will generate a lot of interest in the field and I recommend publishing it.

Reviewer #3 (Remarks to the Author):

The authors have addressed some of my points but I still feel very strong about the title. The title implies more than evidence is presented for. I would suggest an alternative title more in line with the actual data such as "Tetrapod limb and sarcopterygian regeneration share common features".

The other remaining point is the PCR data shown in Fig4c. The authors need to state that the PCRs generate products of the expected size. I only found the quantifications but nothing about that the correct product is actually amplified.

Response to reviewers

We thank the reviewers and editor for comments and suggestions, which have substantially improved our manuscript. Please find below our point-by-point response.

Reviewer #3 (Remarks to the Author):

The authors have addressed some of my points but I still feel very strong about the title. The title implies more than evidence is presented for. I would suggest an alternative title more in line with the actual data such as "Tetrapod limb and sarcopterygian regeneration share common features".

Response: based on the reviewer's concern, we provide below two potential alternative titles. We defer to the editors the decision as to which title to use:

Tetrapod limb and sarcopterygian fin regeneration share a core genetic program

Core components of the tetrapod limb regeneration program predate the fin-limb transition

The other remaining point is the PCR data shown in Fig4c. The authors need to state that the PCRs generate products of the expected size. I only found the quantifications but nothing about that the correct product is actually amplified.

Response: we have now added a sentence on the text stating that the PCR products obtained were at the expected sizes. The actual sizes are described in the pertinent figure legends.